# Cognitive Engagement and Subjective Well-Being in Adults: Exploring the Role of Domain-Specific Need for Cognition

**DOI:** 10.3390/jintelligence12110110

**Published:** 2024-11-03

**Authors:** Julia Grass, Anja Strobel

**Affiliations:** Institute of Psychology, Chemnitz University of Technology, 09107 Chemnitz, Germany; anja.strobel@psychologie.tu-chemnitz.de

**Keywords:** Need for Cognition, investment traits, subjective well-being, life satisfaction, positive affect, negative affect, job satisfaction, study-related satisfaction, emotion regulation, self-control

## Abstract

Need for Cognition (NFC) is an investment trait reflecting interindividual differences in intrinsically motivated engagement in cognitive endeavors. While other motivational constructs like the ability self-concept are typically conceptualized domain-specific, NFC was initially conceived to describe domain-general behavior. Building on a previous study reporting the relevance of domain-specific aspects of NFC in the school context, we investigated the domain-specificity of NFC in predicting various aspects of subjective well-being across three samples with *N* = 1074 participants and subsamples of students and professionals ranging from *n* = 140 to *n* = 346. Our findings reveal positive associations of both domain-general and domain-specific NFC with positive affect, life satisfaction, and job- and study-related satisfaction. In part, domain-specific NFC was more strongly related to domain-specific life satisfaction compared to domain-general NFC. Additionally, we found evidence for small incremental values of domain-specific NFC over and above domain-general NFC in predicting subjective well-being. Confirming previous research, self-control and the emotion regulation strategy reappraisal partially mediated the prediction of well-being by NFC. Our results indicate that additionally considering NFC as domain-specific can meaningfully complement the domain-general conceptualization.

## 1. Introduction

Need for Cognition (NFC) describes relatively stable interindividual differences in approaching cognitively challenging situations and engaging in effortful cognition ([12]). Individuals with higher NFC levels tend to have more elaborated information processing ([13]). NFC has been categorized as being a core investment or intellect trait ([65]), also being related to Curiosity or Openness to Ideas ([50]; [65]), describing “[…] when, where, and how people invest their time and effort in their intellect” ([65]). Hence, NFC is clearly distinguishable from ability, and relations with cognitive abilities are weak to moderate at most, depending on the studied ability and research design ([22]; [36]; [65]).

Defining research on NFC had its origin in social–psychological research on forming judgments and attitudes (for an overview, [13]). In the last four decades, research has produced reliable findings on the nomological network of NFC and on implications of different NFC levels in various contexts (decision making: e.g., [32]; advertising: e.g., [33]; media usage: e.g., [35]; school: e.g., [43]; and health: e.g., [68]). One research topic of increasing interest has been the relation of NFC to various dimensions of human well-being (for an overview, [71]).

### 1.1. Need for Cognition and Subjective Well-Being

Early research indicated negative associations of NFC with depression and anxiety symptoms ([18]; [58]), which were confirmed in later research ([4]; [8]; [52]; [70]). Higher NFC levels have been related to more reported positive life events ([63]) in most studies to increased positive affect (e.g., [22]; [63]; [69]) and sometimes significantly to reduced negative affect (e.g., [69]). Further, NFC was positively associated with subjective evaluations of one’s life (i.e., cognitive subjective well-being; [16]), known as general or domain-specific life satisfaction. Reported associations with global life satisfaction ranged from *r* = −0.01 to *r* = 0.62 with ρ = 0.18 ([71]). Studies on university students reported associations between NFC and study-related well-being ([27], [29], [28]) that are comparable to those between NFC and academic achievement (ρ = 0.20; [46]). All findings indicate that NFC as a cognitive-motivational trait does not only predict intellectual achievements but does also play a relevant role for a broad range of aspects of subjective well-being. Recent meta-analytic findings report a medium overall relationship of ρ = 0.20 between NFC and well-being considering different well-being indicators ([47]; [71]).

### 1.2. Explaining Processes

Different explanations for positive associations of NFC with well-being indicators have been posited theoretically and partly examined. For academic jobs and study-related satisfaction, one line of reasoning is that individuals with higher NFC levels are intrinsically motivated to cope with challenging academic problems (e.g., [27]). Hence, they do not only strive for positive career outcomes but can really enjoy learning, for example, for an exam at university. In general, higher NFC levels are likely to be associated with rather active coping behavior (e.g., [11]), taking flexible perspectives on (decision) situations (for an overview, [13]), goal-oriented behavior (e.g., [22]), and approaching situations perceived as challenging, which can be considered to promote well-being in the long run (e.g., [71]). These explaining approaches follow the idea of personality influencing in what way individuals adapt to certain life events, resulting in different well-being levels ([17]). For positive and negative affect, one longitudinal study provided evidence that differences in NFC are associated with different evaluations of positive and negative life events resulting in differences in affective well-being as follows: Positive life events were experienced more positively, which additionally enhanced positive affective well-being ([63]). All of that can be considered as paths to increased well-being associated with higher NFC levels (for review, see [71]).

To better understand the relations of NFC to well-being, research has to consider potentially underlying processes and mediating variables. Thereby, self-control and emotion regulation strategies refer to processes that predict subjective well-being (e.g., [15]; [31]), were associated with NFC in previous research (e.g., [8]; [29]), and have been shown to mediate associations of NFC with well-being and mental health (e.g., [8]; [52]).

#### 1.2.1. Trait Self-Control

Since Walter Mischel reported results of the popular marshmallow experiment in the 1960s (for review, e.g., [48]), much research has examined the predictive value of individuals’ tendency to resist temptations and to delay gratification for desirable outcomes like academic achievement, life satisfaction, and (mental) health (e.g., [15]; [49]; [64]). Thereby, self-control can be considered as one type of self-regulation, referring to processes to override impulses to resolve a conflict between competing behavioral options (for reviews, [23]; [38]; [66]). From a trait perspective, trait self-control addresses relatively stable individual tendencies to behave in conformity with desired goals in self-control dilemmas, for example, to learn for an exam instead of going out ([14]). Thereby, conceptualizations of trait self-control explain interindividual differences with differences in regulatory abilities and motivational dispositions ([66]). A meta-analysis reported an association of *r* = 0.33 between trait self-control and well-being ([15]). The distinct behaviors that reflect low vs. high trait self-control and lead to the (un)successful resolution of conflicts with relevance for important life outcomes are still discussed ([14]). With NFC referring to cognitive effort investment per definition ([13]) as well as being empirically associated with goal-orientation ([22]) and increased recruitment of resources for cognitively demanding tasks ([51]), higher NFC levels are likely to promote increased self-control. Research on NFC has repeatedly provided evidence for a positive association with trait self-control and a mediating role of trait self-control for predicting well-being indicators by NFC (e.g., [8]; [52]; [72], [71]).

#### 1.2.2. Emotion Regulation Strategies

During the last decades, research on well-being, psychotherapeutic interventions, or coaching has shown increasing interest in the way individuals manage their emotions (e.g., [3]; [60]). Emotion regulation describes different processes modulating the emotional states of individuals, which may be executed consciously but are also often executed without conscious deliberation ([31]). Emotion regulation can be antecedent-focused, referring to processes before emotional responses are fully activated ([31]), or response-focused, referring to response modulation after emotional reactions have already been activated ([31]).

A common example of antecedent-focused strategies is cognitive reappraisal, which describes cognitively reconstruing a situation in a way that changes emotional responses; an example for response-focused strategies is suppression, which refers to modulating the behavioral expression of emotions with inhibiting the expression of an ongoing emotional response ([31]). Concerning well-being and psychopathology, research has provided evidence for beneficial effects of reappraisal and disadvantageous effects of suppression (e.g., [3]; [26]; [31]). This knowledge has been implemented in psychotherapeutic techniques like cognitive restructuring to improve individual well-being (e.g., [30]). Higher NFC is associated with being more open for information deviating from one’s opinion or attitude and to approach cognitive challenges (for an overview, [13]). Being confronted with complex, potentially overwhelming situations can be regarded as both cognitively and emotionally challenging. From a theoretical perspective, approaching thoughts about such situations and adopting different perspectives could be beneficial for well-being compared to avoidance behavior or keeping one (rigid) perspective on a situation. Additionally, higher NFC was associated with self-reporting more positive life events ([63]), possibly resulting from more positively evaluating everyday situations. Hence, higher NFC levels are associated with more flexible perspectives on situations and topics, which in turn should encourage adaptive reappraisal of situations. Those processes may explain why higher NFC levels have been reported to be associated with emotion regulation strategies, especially with cognitive reappraisal ([29]; [37]; [72]). Previous research has identified cognitive reappraisal as a mediating process for the relations of NFC to well-being indicators (e.g., [29]; [37]).

### 1.3. Domain Specificity of Need for Cognition

NFC has been defined as a general behavioral tendency “[…] to engage in and enjoy effortful cognitive endeavors rather than as chronic tendencies toward processing information in particular domains” ([13]). Theoretical approaches to NFC and other investment traits have proposed that investment trait level develops in interaction with intellectual experiences like a cognitively stimulating or rewarding environment during one’s educational biography ([2]; [13]; [65]). [40] ([40]) highlighted that these experiences for children may differ depending on the field of interest or school subject, proposing that domain-specific NFC profiles may exist additionally to domain-general NFC. Their investigation on ninth-graders provided evidence for the meaningfulness and validity of domain-specific aspects of NFC in addition to domain-general NFC in the context of subject-specific school achievements ([40]).

Following the symmetry principle for hierarchical constructs and the idea of bandwidth-fidelity (e.g., [41]), research on implications of NFC for well-being could profit from a domain-specific perspective on NFC because the strength of associations probably depends on the fit of construct levels. Subjective well-being refers to subjective evaluations of life both in general and specific domains (e.g., job satisfaction). Based on [40] ([40]) and the abovementioned symmetry principle of constructs, conceptualizing NFC in adults also domain-specifically could improve realistic estimations of its role, especially for domain-specific well-being measures.

### 1.4. Present Research

Based on previous research on schoolchildren ([40]), we examined the value of conceptualizing NFC in adults not only domain-generally but also domain-specifically in the context of subjective well-being using data of three samples recruited between 2020 and 2023. Based on previous research providing evidence for the predictive value of NFC for well-being (for a review, [71]), we assumed at least medium positive associations with positive affect (H1) and low negative associations with negative affect (H2), as well as medium positive associations between NFC and global life satisfaction (H3). For domain-specific life satisfaction in students, we assumed increasing NFC to be associated with increasing study-related satisfaction, expecting strongest associations with subject-related satisfaction based on previous research (H4a; [28]). For employed individuals, we assumed a positive association between NFC and job satisfaction (H4b). Taking a domain-specific perspective on NFC comparable to the study by [40] ([40]), we assumed incremental predictive values over and above general NFC of study-specific NFC for predicting study-related satisfaction and of job-specific NFC for predicting job satisfaction (H5). We assumed trait self-control and cognitive reappraisal as partly explaining processes and examined them as mediators for significant associations of NFC with well-being indicators (H6). Data collections of Samples 2 and 3 aimed at replicating the first data collection and at providing an increased robustness of findings and conclusions.

## 2. Material and Methods

In this article, we report about different data collections at three time periods between June 2020 and February 2023. We report how we determined our sample size, all data exclusions, all manipulations, and all measures in this study. All data collections have been preregistered. Preregistrations are available under https://aspredicted.org/95pq-hyhb.pdf (accessed on 28 October 2024; Sample 1), https://aspredicted.org/2tn4-xtgv.pdf (accessed on 28 October 2024; Sample 2), and https://aspredicted.org/2swy-sd7s.pdf (accessed on 28 October 2024; Sample 3). The procedure was ethically approved by the local ethics committee (reference: #101551806). Raw data and SPSS syntax are available at https://doi.org/10.17605/osf.io/y6evn (accessed on 28 October 2024).

### 2.1. Procedure

The data of all three samples were collected via German online surveys of similar structure and very comparable content. For Sample 1, we used the Software EFS Survey ([56]). Due to updated requirements of our institution, all questionnaires were provided with [44] ([44], [45]) for Samples 2 and 3. At the beginning, all participants had to give their informed consent for participation. First questions referred to demographic information followed by questionnaires assessing the psychological constructs of interest: domain-general NFC, emotion regulation, self-control, affective well-being, global life satisfaction, domain-specific NFC, and domain-specific life satisfaction. Domain-specific questionnaires referred to study-specific NFC and study-related satisfaction for students and to job-specific NFC and job satisfaction for employed participants. They were provided only for participants that stated to currently study at university or to be employed, respectively. Finally, control variables were assessed, and information about options for compensation was provided. Participants studying psychology or a related subject at the university affiliated with the first author could receive study credit for compensation. All participants of Samples 1 and 2 could take part in a raffle to win 10, 20, or 50 euros. All valid participations in Sample 3 were compensated with 3.45 euros via prolific.co. For Sample 3, we followed requirements of the platform prolific.co and included two attention checks following the platform’s policy ([55]).

### 2.2. Participants

All three samples were recruited in Germany. Following the recommendation for robust correlation coefficients of [61] ([61]), a sample size of 250 was aimed at for all samples. The main characteristics of all samples are displayed in Table 1. For Samples 1 and 2, participants were recruited via social platforms, mailing lists, and private contacts. Data collection for Sample 1 took place during the first COVID-19 wave, so we used mainly online recruiting methods. For Sample 2, we used additionally printed flyers in public spaces. Sample 3 was recruited via prolific.co. Samples 2 and 3 aimed at replicating the results of Sample 1 to provide a valid basis for conclusions about the value of domain-specific conceptualizations of NFC in the context of subjective well-being.

**Sample 1.** The first dataset was collected between 17 June and 30 July 2020, and 549 individuals provided their consent to participate. Of them, 460 finalized all relevant questions, including a statement about their honesty of responses. We excluded participations when participants indicated answering dishonestly or without reading the questions, stated technical problems, or stated to currently suffer from a severe psychiatric disorder. The final sample consisted of *N*_1_ = 451 participants who were highly educated, with 92% having a university entrance qualification or university degree. The student subsample consisted of *n*_1*s*_ = 258 participants, and the employed subsample of *n*_1*w*_ = 159 participants, ranging between full-time jobs and part-time jobs less than 15 h a week. Of the student subsample, 34.7% (*n* = 90) studied psychology or a closely related subject. For the subsample of employed participants, the most frequent fields of occupation were social and cultural services (23.9%), health (18.2%), and business management and organization (8.8%). Effect size sensitivity was calculated post hoc with G*Power ([19]). It identified ρ_N1_ ≥ 0.11, ρ_n1s_ ≥ 0.16, and ρ_n1w_ ≥ 0.20 as minimum population effect sizes detectable with 1 − β = 0.80 and α = 0.05 (one-tailed).

**Sample 2.** In the second data collection, from 30 May to 26 June 2022, 435 individuals agreed to the conditions of participation. Of them, 340 finalized all relevant questions, including a statement about their honesty of responses. Three participants were excluded because they stated to have answered dishonestly or without reading the questions. Two participants were excluded due to their information about suffering currently from severe psychiatric symptoms (psychiatric comorbidities for years, one person with recent inpatient treatment). Hence, the final sample size was *N*_2_ = 335 with *n*_2*s*_ = 157 students and an employed subsample of *n*_2*w*_ = 156 with participants working between full-time jobs and part-time jobs less than 15 h a week. Participants were highly educated, with 85.1% having a university entrance qualification or university degree. For the employed subsample, the most frequent fields of occupation were social and cultural services (17.3%), health (12.8%), business-related service (12.8%), construction (11.5%), and IT and science (9.6%). Of the student subsample, 54.1% (*n* = 85) studied psychology or a closely related subject. The data were collected after two years of the COVID-19 pandemic and three months after Russia invaded Ukraine. In total, 51.6% of participants stated to feel more burdened, 46.9% to worry more than usual. Effect size sensitivity was calculated post hoc with G*Power ([19]). It identified ρ_N2_ ≥ 0.14 and ρ_n2s/n2w_ ≥ 0.20 as minimum population effect sizes detectable (1 − β = 0.80, α = 0.05, one-tailed).

**Sample 3.** Because the sizes of employed subsamples in Samples 1 and 2 were smaller than a priori analyses specified, the third data collection aimed only at employed participants. The survey was conducted between 20 January and 9 February 2023. Hence, for the recruitment via prolific corresponding inclusion/exclusion criteria were selected out of different criteria provided on the platform for sample selection: part-time or full-time employed with at least 20 h/week and fluency in German language at C1 level. An a priori power analysis ([19]) showed that 346 subjects would be needed to detect a small effect of *r* = 0.15 (α = 0.05, 1 − β = 0.80, two-tailed). Hence, 346 participants were the target sample size requested from Prolific. After 346 participations were completed, inclusion criteria were checked by answers to demographic questions in our survey. Five participants were excluded because they did not match the abovementioned selection criteria. Three participants were excluded because they failed attention checks, stated not having read questions completely, or were exceptionally fast. For those rejected participations, eight participants were additionally recruited so that the final sample consisted of *N*_3_ = 346 data sets. The most frequent fields of occupation were IT and science (31.8%), business-related service (16.8%), social and cultural services (14.2%), and health (8.4%). The final sample was highly educated, with 88.7% having a university entrance qualification or university degree. Similar to Sample 2, participants were asked whether they worry more or feel more burdened than usual. About one-third of participants agreed to at least one statement (worries: 29.5% and burdened: 31.8%).

### 2.3. Materials and Methods

#### 2.3.1. Need for Cognition

The assessment of NFC was based on the study by [40] ([40]). Domain-general NFC was assessed with five German items, for example, “I like situations that require me to think hard” ([40]). Participants had to answer items on a 7-point scale ranging from −3 (*completely disagree*) to +3 (*completely agree*). Answers were aggregated as sum scores, with higher scores indicating higher NFC levels, theoretically ranging from −15 to +15.

Domain-specific NFC was assessed with five items based on [40] ([40]) and adapted to work and study contexts, respectively. Table 2 displays all items. Response format and scoring were similar to domain-general NFC. In all samples, domain-specific NFC correlated strongly with domain-general NFC (0.71 ≤ *r_s_* ≤ 0.82, *p* < 0.001).

#### 2.3.2. Domain-General and Domain-Specific Life Satisfaction

For the assessment of global life satisfaction, we used a German version of the Satisfaction with Life Scale ([25]). It consists of five items (e.g., “In most ways, my life is close to my ideal”) rated on a 7-point scale ranging from 1 (*completely disagree*) to 7 (*completely agree*). Individual scores were the sum of item responses.

Job satisfaction was assessed with a German short scale consisting of eight items like “I’m really enjoying my work” ([21]). Items were rated on a 5-point scale whose labels differed depending on the respective item (e.g., 1 = *wrong* to 5 = *true*). All answers were summed up.

We assessed study-related satisfaction with a German nine-item short scale ([67]) representing three dimensions: satisfaction with study conditions, coping with study-related stress, and subject-related satisfaction (e.g., “I really enjoy what I study”). Participants had to answer on an 11-point rating scale ranging from 0 (*completely disagree*) to 100 (*completely agree*). Inverted items were recoded, and answers were averaged per dimension so that higher scores indicated higher study-related satisfaction for every dimension.

#### 2.3.3. Affective Well-Being

Positive and negative affect were assessed with the German version of the Positive and Negative Affect Schedule ([39]). It consists of 10 adjectives describing positive affect (e.g., “active”) and 10 adjectives describing negative affect (e.g., “distressed”). Participants rated each adjective on a 5-point rating scale (1 = *not at all* to 5 = *very*), referring to how they feel in general. Scores were calculated separately for positive and negative affect as sums of item responses.

#### 2.3.4. Emotion Regulation Strategies

Emotion regulation was assessed with a German version of the Regulation Questionnaire ([1]) with six items assessing cognitive reappraisal (e.g., changing thoughts for more positive emotions) and four items assessing suppression (e.g., not expressing emotions). Item responses referred to a 7-point Likert scale ranging from 1 (*not true at all*) to 7 (*completely true*). Scores were calculated for each type of strategy as averaged item responses.

#### 2.3.5. Trait Self-Control

We assessed trait self-control with a German 13-item questionnaire ([7]). Items represent behavior indicating dispositional self-control capacity (e.g., ability to work effectively toward long-term goals) and were rated on a 5-point scale from 1 (*completely disagree*) to 5 (*completely agree*). Item responses were averaged.

Table 3 presents descriptive statistics and reliability for all implemented instruments.

#### 2.3.6. Further Variables

We assessed demographic data and information about living situation, job situation, or subject and duration of study. The assessment of further variables differed slightly between the three samples due to different social conditions at the time of each data collection. As the surveys of Sample 1 and Sample 2 were conducted when the COVID-19 pandemic had stronger influence on everyday life and potentially on well-being, too, we added items focusing on COVID-19-related conditions (https://osf.io/y6evn/?view_only=4ae7e4377a5546ed8f7c9ae3dc527292, accessed on 28 October 2024). In Sample 3, two more general questions were asked as follows: whether individuals currently feel more burdened than usual and whether they worry more than usual.

In all samples, two questions referring to data validity were placed at the end of the survey, referring to honesty of responses and reading all questions before answering.

### 2.4. Statistical Analysis

All analyses were conducted with IBM SPSS Statistics (version 29). The distribution of variables of interest was tested with QQ-Plots, histograms, and Kolmogorov–Smirnov tests. Due to deviations from normality, Spearman rank coefficients were used for correlation analyses. For hypothesis testing, we applied the Bonferroni correction to adjust the significance level by considering 80 single comparisons (α = 0.05/80), resulting in α ≈ 0.001. Correlations were interpreted following the recommendations by [24] ([24]).

Incremental predictive value was tested using hierarchical regression analyses using robust analyses based on 1000 bootstrap samples with 95% bias-corrected and accelerated (BCa) confidence intervals. In a first step, age or gender were included in the regression analyses if correlated with the respective criterion with *p* ≤ 0.05. Due to low frequencies of participants choosing “diverse” for gender (per subsample, *n* = 0–2), point-biserial correlations as well as regression analyses, including gender, were calculated excluding diverse participants. Gender and age were not generally included in all analyses because results would change very unlikely if no correlations were observed, and generally including them would shift the focus to demographic variables that are not theoretically driven relevant for the current research. Second, domain-general NFC was included before domain-specific NFC related to one’s job or studies.

For estimating overall effects of domain-specific NFC across all samples, we analyzed multilevel models with data collection (i.e., Sample 1, Sample 2, and Sample 3) and domain (studies vs. work) as random factors. Multilevel analyses were conducted with RStudio (version 2023.12.0; [54]) using R (Version 4.2.1; [57]) and the packages lme4 ([6]) and lmerTest ([42]). Mediation analyses were calculated for the prediction of all well-being indicators NFC was associated with. Analyses were performed with PROCESS (version 4.0; [34]), calculating 95% BCa confidence intervals based on 1000 bootstrap samples. Significant effects were indicated by confidence intervals excluding zero.

## 3. Results

### 3.1. Associations of Need for Cognition with Subjective Well-Being

Results of correlation analyses with NFC are displayed in Table 4 and Table 5. Intercorrelations of all well-being variables, emotion regulation, and self-control are uploaded in the electronic Appendix A (see Appendix A).

In all samples, both domain-general and domain-specific NFC was strongest associated with positive affect (0.25 ≤ *r_s_* ≤ 0.39). Negative affect was not associated with NFC. In most (sub)samples, NFC was weakly associated with life satisfaction, which was strongest for job-specific NFC (0.21 ≤ *r_s_* ≤ 0.24). Associations of NFC with domain-specific life satisfaction were strongest for job satisfaction and subject-related satisfaction with one’s studies (0.16 ≤ *r_s_* ≤ 0.36). NFC was not associated with satisfaction with study conditions and only in Sample 2 weakly with satisfaction related to coping with study-related stress.

The sizes of correlations of NFC with cognitive reappraisal and self-control were heterogenous, being very small and non-significant in some sub(samples) and moderate in others (e.g., with self-control *r_s_* = 0.20/0.26 in Sample 3, with reappraisal 0.16 ≤ *r_s_* ≤ 0.22 in Sample 1). NFC did not correlate with suppression.

### 3.2. Incremental Validity of Domain-Specific Need for Cognition

For all well-being indicators that were significantly correlated with domain-specific or domain-general NFC, hierarchical regression analyses were conducted to identify the incremental value of domain-specific NFC. Additionally, multilevel models considering time of data collection (i.e., Sample 1 to Sample 3) and domain (study-related vs. job-related) as random factors (random intercept) were calculated. Similar to hierarchical regression analyses, multilevel models were calculated for well-being indicators correlated with NFC in at least one sample. Due to sample-dependent (non-significant) effects of gender and age together with differences in assessing age in Sample 1 compared to Samples 2 and 3, control variables were not considered in these multilevel analyses. We calculated two models for predicting each criterion by including general NFC first and domain-specific NFC second to calculate ∆*R*^2^. For domain-specific well-being (i.e., study-related and job satisfaction), we calculated separate analyses in the overall working sample and the overall study sample, respectively. In these analyses, we used only data collection as a random factor because domain-specific well-being assessments were not as comparable as necessary to define them as two domain-specific aspects of the same factor.

#### 3.2.1. Positive Affect

Results of sample-specific regression analyses for positive affect are displayed in Table 6. In both student samples, general NFC predicted positive affect with β = 0.17/0.32 (*p* = 0.005, *p* < 0.001). Adding study-specific NFC explained no significant additional variance (*p* = 0.100/0.121). In employed subsamples, general NFC predicted positive affect with 0.30 ≤ β ≤ 0.35 (*p* < 0.001). Job-specific NFC had a small incremental predictive value (0.018 ≤ ∆*R*^2^ ≤ 0.037, 0.24 ≤ β ≤ 0.29, 0.008 ≤ *p* ≤ 0.015).

Results of random effect models to calculate an overall effect across all samples and both types of domain-specificity of NFC (i.e., study-specific and job-specific) are displayed in Table A1. They show a small incremental value of domain-specific NFC with ∆*R*^2^*_marginal_* = 0.024 and β = 0.22 [0.13, 0.30] over and above domain-general NFC. The predictive value of domain-general NFC decreased from β = 0.31 to β = 0.15 after including domain-specific NFC, which indicates redundancy effects. Together, domain-general and domain-specific NFC explained 12% of variance in positive affect (*R*^2^*_marginal_* = 0.115).

#### 3.2.2. Life Satisfaction

Table 7 displays sample-specific hierarchical regression analyses for life satisfaction. 

In all samples, domain-specific NFC incrementally predicted life satisfaction over and above general NFC (study-specific NFC: ∆*R*^2^ = 0.022/0.041, β = 0.20/0.26, *p* = 0.017/0.010; job-specific NFC: 0.021 ≤ ∆*R*^2^ ≤ 0.027, 0.22 ≤ β ≤ 0.27, 0.006 ≤ *p ≤* 0.044).

Multilevel results for identifying an overall effect across all samples and both study- and job-specific NFC are displayed in Table A2. The overall predictive value of domain-specific NFC for predicting life satisfaction was β = 24 [0.16, 0.33]. The incremental predictive value over and above domain-general NFC was small, with ∆*R*^2^*_marginal_* = 0.026. The predictive value of domain-general NFC decreased from β = 0.11 to β = −0.06 (n.s.) after including domain-specific NFC, indicating redundancy effects. Together, domain-general and domain-specific NFC explained 4% of variance in life satisfaction (*R*^2^*_marginal_* = 0.039).

#### 3.2.3. Study-Related Satisfaction

We analyzed the incremental value of study-specific NFC for predicting subject-related satisfaction in both samples and for predicting satisfaction related to coping with study-related stress in Sample 2, corresponding with significant correlational findings for dimensions of study-related satisfaction (see Table 8).

In the first step, study-related satisfaction was predicted by general NFC only in Sample 1, referring to subject-related satisfaction with β = 0.24, *p* < 0.001. Further, study-specific NFC had an incremental value for predicting subject-related satisfaction in both samples (∆*R*^2^ = 0.082/0.071, β = 0.39/0.35, *p* < 0.001) and coping with study-related stress in Sample 2 (∆*R*^2^ = 0.033, β = 0.24, *p* = 0.022).

Results of random effect models to calculate an overall effect across all samples are displayed in Table A3. The overall incremental value of study-specific NFC was ∆*R*^2^*_marginal_* = 0.079 for predicting subject-related satisfaction (β = 0.38 [0.26, 0.50]) and ∆*R*^2^*_marginal_* = 0.016 for predicting satisfaction referring to study-related coping (β = 0.17 [0.04, 0.30]). Together, domain-general and study-specific NFC explained 12% of variance in subject-related satisfaction (*R*^2^*_marginal_* = 0.117) and 2% of variance in satisfaction related to coping with study-related stress (*R*^2^*_marginal_* = 0.021). For subject-related satisfaction, redundancy effects were indicated by β decreasing from 0.20 to −0.06 (n.s.) after including study-specific NFC.

#### 3.2.4. Job Satisfaction

The results of regression analyses predicting job satisfaction are provided in Table 9. In Samples 1 and 3, job-specific NFC incrementally predicted job satisfaction over and above general NFC (∆*R*^2^ = 0.07, β = 0.38/0.48, *p* < 0.001). In Sample 2, general NFC explained 7.7% of variance with no incremental value of job-specific NFC (∆*R*^2^ = 0.007, β = 0.15, *p* = 0.278).

Results of random effect models to calculate an overall effect across all samples are displayed in Table A4. The overall incremental predictive value of job-specific NFC for predicting job satisfaction over and above domain-general NFC was ∆*R*^2^*_marginal_* = 0.047 with β = 0.36 [0.24, 0.47]. Redundancy effects were indicated for domain-general NFC with a decreased β from 0.16 to −0.12 after including job-specific NFC. Together, domain-general and job-specific NFC explained 7% of variance in job satisfaction (*R*^2^*_marginal_* = 0.072).

### 3.3. Mediation Analyses

To examine the role of self-regulatory processes for predicting well-being by NFC, we ran separate mediation analyses in each sample using emotion regulation strategies and self-control as parallel mediators (see Table 10). Analyses were calculated using general or domain-specific NFC as a predictor and a well-being indicator as a criterion. As predicted criteria, we used all well-being indicators except negative affect and satisfaction with study conditions because both had no observable associations with NFC.

Except for two subsamples, respectively, we found evidence for a mediating role of emotion regulation and/or self-control for the prediction of positive affect and life satisfaction by NFC. Thereby, for predicting positive affect, a direct effect of NFC remained in all cases. For predicting life satisfaction, a direct effect remained only for job-specific NFC in Sample 3, with an existing indirect effect through regulatory processes. The other mediation effects for predicting life satisfaction were complete mediations.

Results for domain-specific satisfaction related to one’s studies or job were more ambiguous. Study-related satisfaction referring to coping was not predicted indirectly by NFC but directly by study-specific NFC in Sample 2. Satisfaction with one’s study subject was predicted by study-specific NFC directly in Sample 1 and Sample 2, with an additional indirect effect in Sample 2. For job satisfaction, emotion regulation and self-control had mediating effects in half of the (sub)samples. In the other cases, (domain-specific) NFC predicted job satisfaction without mediating regulatory processes. Only in Sample 1, the effect of job-specific NFC was completely mediated. In general, significant mediating paths mostly refer to reappraisal and self-control.

## 4. Discussion

The current research had two aims. We investigated the potential advantages of considering the NFC domain-specifically in addition to the common domain-general conceptualization in the context of subjective well-being. Furthermore, we examined the mediating role of emotion regulation strategies and self-control as explaining processes.

Our results are based on three data sets, including subsamples of students and employed participants, with a total sample size of *N* = 1132. As expected, domain-specific and domain-general NFC were highly correlated. Descriptively, NFC levels were higher in working compared to student subsamples. Compared to domain-general NFC, domain-specific NFC was about 0.5 SD reduced in students. We can only speculate about the reasons for those differences. One reason for lower study-specific NFC levels compared to domain-general NFC might be that study subjects cover a broader range of content that may be of less interest for some students, so that study-related effort is likely to be both sometimes intrinsically motivated and related to NFC as well as sometimes extrinsically motivated. Further, opportunities to enact higher NFC levels when learning for exams and dealing with study content are likely to depend on study curricula, which may have influenced answering study-specific NFC items. Furthermore, the reduction in study-specific NFC compared to domain-general NFC may have been associated with adverse study conditions with long periods of home studying instead of regularly studying on campus. Although NFC levels above the theoretical mean conform with previous findings in students, we would not have expected similar and even higher NFC levels for working participants. Prospective studies should examine whether our NFC levels above the theoretical mean are representative for working individuals or mirror selection bias.

### 4.1. Associations Between NFC and Subjective Well-Being

Consistently in all samples, we found moderate associations of domain-general and domain-specific NFC with positive affect (0.25 ≤ *r_s_* ≤ 0.39), which was in line with a meta-analytically reported moderate association of ρ = 0.20 ([71]). Also in line with previous results, we found no association with negative affect and satisfaction with study conditions ([29], [28]; [71]). Similar to previous meta-analytic findings ([71]), associations with life satisfaction were small to moderate, ranging between 0.09 ≤ *r_s_* ≤ 0.24 and being strongest for job-specific NFC.

Job satisfaction was weakly to moderately associated with domain-general NFC (0.11 ≤ *r_s_* ≤ 0.26). Associations with job-specific NFC were at least medium and descriptively stronger (0.25 ≤ *r_s_* ≤ 0.33). Subject-related satisfaction with one’s studies correlated also weakly to moderately with domain-general NFC (0.16 ≤ *r_s_* ≤ 0.31) as well as with study-specific NFC (0.25 ≤ *r_s_* ≤ 0.36). Again, associations with domain-specific NFC were descriptively stronger than with domain-general NFC. We found small to medium associations (*p* < 0.05) between both domain-general and domain-specific NFC, with satisfaction referring to coping with study-related stress only in Sample 2 (*r_s_* = 0.16/0.22). That is, both domain-general and domain-specific NFC showed comparable correlational patterns in all samples.

Our results provide cautious evidence for stronger associations between domain-specific life satisfaction and domain-specific NFC compared to domain-general NFC. Global life satisfaction was also strongly associated with job-specific NFC, indicating that NFC in the job context can be relevant not only for domain-specific life satisfaction.

### 4.2. Incremental Predictive Value of Domain-Specific Need for Cognition

Inspired by a previous study on children ([40]), we examined the meaningfulness of a domain-specific conceptualization of NFC in the sense of incremental predictive validity for subjective well-being criteria. Subjective well-being refers to emotional experience in the sense of positive or negative affect and global or domain-specific evaluations of one’s life (e.g., [10]; [16]). We thereby examined the incremental value of domain-specific NFC for predicting both domain-general and domain-specific well-being (except for negative affect, which was uncorrelated with NFC).

Corresponding with at least moderate associations of (domain-specific) NFC with general positive affect, domain-general NFC explained a small to moderate amount of variance in all samples (overall *R*^2^ = 0.091 across all samples). The overall incremental value of domain-specific NFC across domains and samples was ∆*R*^2^ = 0.024. Although study-specific NFC had no incremental predictive value over and above domain-general NFC, job-specific NFC explained small incremental variance in all three samples. Hence, the prediction of positive affect was improved by including job-specific NFC for employed participants.

Only in two out of five subsamples, domain-general NFC had a small, significant predictive value for predicting life satisfaction. Adding study-specific or job-specific NFC in a further step incrementally improved the prediction of life satisfaction in all subsamples. The overall incremental value of domain-specific NFC for predicting life satisfaction across samples and domains was ∆*R*^2^ = 0.026. Hence, for the prediction of life satisfaction, domain-specific NFC was equally to even more important than domain-general NFC. This finding was partly surprising because, following the Brunswik symmetry principle (e.g., [41]), we would have assumed general NFC to be more meaningful for the prediction of global life satisfaction compared to domain-specific NFC. One possible explanation refers to the idea that individuals may think of specific life contexts as anchoring references when asked to evaluate their life satisfaction in general.

That is supported by moderate to strong associations of both job satisfaction and study-subject-related satisfaction with global life satisfaction in our samples (0.36 ≤ *r_s_* ≤ 0.48). Enjoying study-related or job-related cognitive activities in the sense of domain-specific NFC may be of relevance for life satisfaction, especially when studying or working is a large part of everyday life. The latter was probably true for most employed participants because most of them stated to work full-time. For students, we assume that the relevance of studies for individual evaluations of life is more heterogeneous. To gain more insight into possible explanations for the predictive value of domain-specific NFC for life satisfaction, future studies could further examine the frame of reference when individuals rate global life satisfaction. We also cannot exclude situation-specific influences on our results since data collections for Samples 1 and 2 were conducted during the COVID-19 pandemic, which changed life circumstances, including studies and job conditions at that time meaningfully.

Remarkably, the explained variance proportions by domain-general and domain-specific NFC were quite different comparing positive affect and life satisfaction with *R*^2^*_marginal_* = 0.12 for positive affect and *R*^2^*_marginal_* = 0.04 for life satisfaction when controlling for variance explained by the random factors domain and time of data collection. Hence, positive affect could be much better predicted by domain-specific and domain-general NFC than a cognitive evaluation in the sense of life satisfaction, which underlines the relevance of NFC as a resource for affective outcomes and emotional adaptation ([8]; [28]; [71]).

While general NFC predicted subject-related satisfaction with one’s studies only in Sample 1, study-specific NFC had an incremental predictive value in all conducted regressions: Subject-related satisfaction with one’s studies was incrementally predicted by study-specific NFC in both samples. Study-specific NFC explained small to medium variance proportions with overall ∆*R*^2^ = 0.079 and β = 0.38 [0.26, 0.50] across both samples. In Sample 2, also satisfaction referring to coping with study-related stress was incrementally predicted by study-specific NFC. Overall, NFC explained a much smaller variance proportion in coping-related satisfaction (*R*^2^ = 0.021) compared to subject-related satisfaction (*R*^2^ = 0.117). The findings confirmed our hypothesis of study-specific NFC adding predictive value, especially for the prediction of subject-related satisfaction with one’s studies. They also highlight that study-specific NFC, though strongly correlated to general NFC, is not completely redundant to it. Instead, it covers unique facets of individual behavior for predicting some criteria.

Two of three analyses regarding job satisfaction indicated an incremental value of job-specific NFC over and above general NFC. The random effect model across all samples identified an overall effect of ∆*R*^2^ = 0.047 with β = 0.36 [0.24, 47]. This finding further supports the idea of a meaningful conceptualization of NFC both domain-generally and domain-specifically ([40]).

### 4.3. Mediating Processes

For adding to insights in processes underlying implications of NFC on subjective well-being, we built on previous research (e.g., [8]; [28]; [52]; [72]) and examined emotion regulation and trait self-control as mediating variables. Confirming previous research, NFC was at least weakly positively related to cognitive reappraisal and self-control and not associated with emotion suppression in most subsamples. Matching this result, indirect effects of NFC in the prediction of well-being indicators occurred mainly via self-control or reappraisal. Mediation analyses provided evidence for a mediating role of reappraisal and self-control, mainly for the prediction of positive affect, global life satisfaction, and, in part, job satisfaction. In many cases, a direct effect of NFC remained. Overlapping confidence intervals indicated that indirect effects of domain-specific and domain-general NFC were comparable in size. Altogether, the results provided evidence for an at least partial mediating role of trait self-control and/or reappraisal for the prediction of positive affect, job-specific, and global life satisfaction. That finding confirms previous research identifying NFC as a resource for (emotional) challenges in life and self-regulatory processes (e.g., [8]; [11]; [62]).

In some subsamples, associations between NFC and job-specific or global life satisfaction were completely attributable to differences in self-control and emotion regulation, whereas the association between positive affect and NFC was only in part attributable to self-control and reappraisal in all samples. Individuals with higher NFC levels may not only report increased positive affect because they tend to be more long-term oriented and adapt helpful perspectives. Additional possible explanations are, for example, that they tend to approach challenges that can reduce anxiety and avoidance behavior or that they tend to really enjoy potentially exhausting and strenuous cognitive tasks, which are likely to be part of one’s studies and jobs following higher education, as most participants had. In general, the remaining direct effects of NFC indicated more complex explanatory paths for associations with subjective well-being. Between-sample differences in effect sizes of (in)direct effects suggest that future studies should investigate possible moderating variables.

### 4.4. Limitations

To allow for increased generalizability of results, we examined our research questions in different samples and collected data at different times. Although we used different methods for recruitment, all three samples were highly educated, which may have led to variance restriction and reduced generalizability. To reach not only highly educated individuals, future studies should use platforms like we used in Sample 3 and choose selection criteria that aim at low to middle education levels for more balanced proportions of different education levels in samples. However, even in Sample 3, NFC values were above the theoretical mean of 0, and higher NFC values may be confounded with increased willingness to participate due to increased scientific interest ([20]). Higher education levels among survey participants are likely to be found due to more familiarity with science and respective self-selection bias. Furthermore, especially Samples 1 and 2 participated during a challenging time of a worldwide pandemic (COVID-19) and the beginning of the Russia–Ukraine war that had negative implications for well-being in European countries (e.g., [53]; [59]). On the one hand, this limits generalizability. On the other hand, the war is still ongoing, other political and social challenges like climate change can be important situational influences on well-being ([5]), and results for most well-being indicators were quite stable among subsamples and comparable to previous results. That is, we assume mean level changes in well-being due to crises like the COVID-19 pandemic are not necessarily changing correlations of well-being with NFC, which is indicated by results of multilevel analyses.

For assessing NFC domain-specifically, we used items closely based on [40] ([40]), which have been developed for the purpose of assessing NFC domain-specifically but have not been used in many studies yet. Consequently, broad evidence for its validity is still lacking for domain-specific NFC items, and our results are promising but preliminary, as described in [40] ([40]). Estimates of reliability (i.e., Cronbach’s α) were very good, and our results with quite comparable patterns to general NFC argue for the validity of both scales. Future research should continue to use these items and provide further indicators of their psychometric quality. Questionnaires to assess NFC domain-specifically and domain-generally were presented always in the same order with domain-general items first with the idea that participants would pay more attention to the domain-specific character when having answered similar items without domain-specific context compared to the opposite order. The downside of that decision are possible sequence effects for completing the questionnaires referring to domain-general and domain-specific NFC comprising very similar items. As expected, domain-specific and domain-general NFC were highly correlated, and due to probable multicollinearity, regression weights should be interpreted carefully when both are included. Our research questions have focused on examining the incremental value of domain-specific NFC. Consequently, our conclusions did not rely on regression weights but mainly considered changes in explained variance in models including both domain-specific and domain-general NFC compared to models including only domain-general NFC.

Online surveys are very common to efficiently assess individuals in different regions and to lower the threshold for participation. However, they entail the risk that we were not able to ensure comparable conditions and an undisturbed environment during responding for the participants, so that conditions of participation may have differed and may have affected the validity of our results. Anonymous participation should encourage honest responses, but we cannot rule out socially desirable response patterns completely.

The cross-sectional design of our study does not allow for conclusions on causal relations between NFC and subjective well-being, so unidirectional theoretical explanations for associations between NFC and well-being could not be explicitly tested with our data. Hence, using NFC as a predictor and well-being indicators as criteria in regression and mediation analyses followed theoretical assumptions (for an overview, see also [71]) and only one directional path of a probably reciprocal relation. Examining causality with longitudinal designs should be part of future research. For example, one previous study on depressive patients indicated NFC as “a personal capacity that could not only favor the change in self-regulation as a consequence of decreasing depressive symptomatology but rather serve as an independent and at least partial catalyst for this change” ([62]). For research on reciprocal paths, including effects of well-being on NFC, prospective studies should also consider cognitive motivation from a state perspective ([9]) instead of focusing on its trait aspects as we did. Examining causal pathways between NFC and well-being with longitudinal designs should be part of future research.

## 5. Conclusions

The current research provides encouraging evidence that assessing NFC domain-specifically meaningfully complements its domain-general conceptualization. Domain-specific and domain-general NFC were strongly associated. We found comparable correlation patterns of domain-specific and domain-general NFC with well-being indicators, self-control, and emotion regulation. Especially life satisfaction, both domain-general and domain-specific, was partly strongly associated with domain-specific NFC. A domain-specific perspective incrementally increased explained variances of study-related and job satisfaction. For employed individuals, job-specific NFC additionally had an incremental value for the prediction of domain-general well-being in the sense of positive affect and global life satisfaction. Further, our research confirmed previous research that reported self-control and the emotion regulation strategy reappraisal as partially mediating variables for the prediction of well-being by NFC. Our results provide further evidence for NFC being a resource for subjective well-being and suggest that domain-specific aspects of NFC can relevantly add to our understanding of the implications of interindividual differences in NFC.

## Figures and Tables

**Table 1 jintelligence-12-00110-t001:** Demographic data of (sub)samples.

	Sample 1	Sample 2	Sample 3
*N*	451	335	346
Age	*Mdn* = 21–29 years ^a^	*M* = 30.75 ± 12.28 years	*M* = 32.4 ± 8.5 years
Gender	73.4% female 26.6% male	72.5% female 27.2% male 0.3% diverse	32.7% female 66.8% male 0.6% diverse
Student Subsamples			
*N*	258	157	-
Age	*Mdn* = 21–29 years ^a^	*M* = 23.03 ± 3.35 years	-
Gender	76.0% female 24.0% male	79.0% female 20.4% male 0.6% diverse	- -
Employed (Sub)samples			
*N*	159	156	346
Age	*Mdn* = 30–39 years ^a^	*M* = 37.2 ± 12.2 years	32.4 ± 8.5 years
Gender	69.2% female 30.8% male	64.1% female 35.9% male	32.7% female 66.8% male 0.6% diverse
Work status	66.0% full-time	78.2% full-time	81.5% full-time

*Note.* ^a^ In Sample 1, age was assessed as an ordinal variable with age categories.

**Table 2 jintelligence-12-00110-t002:** Items for domain-specific Need for Cognition.

German	English ^a^
1. *Beim Arbeiten [im Studium]* löse ich gerne Aufgaben, bei denen man richtig nachdenken muss.	*At work [at university]*, I like solving problems that require hard thinking.
2. *Beim Arbeiten [im Studium]* mag ich Situationen, in denen ich richtig nachdenken muss.	*At work [at university]*, I like situations that require me to think hard.
3. Wenn ich *beim Arbeiten [im Studium]* Aufgaben zum Nachdenken bekomme, dann freue ich mich.	*At work [at university]*, I’m happy when I get an assignment that requires me to think hard.
4. *Beim Arbeiten [im Studium]* denke ich sehr gerne nach.	*At work [at university]*, I like to think a lot.
5. *Beim Arbeiten [im Studium]* macht mir Nachdenken Spaß.	*At work [at university]*, thinking is fun for me.

*Note*. Text in italics was used for assessing job-specific NFC; text in brackets for study-specific NFC. ^a^ Not an official translation, and this is only provided for this article.

**Table 3 jintelligence-12-00110-t003:** Descriptive statistics of used instruments.

	*M*			*SD*			Cronbach’s α		
	S1	S2	S3	S1	S2	S3	S1	S2	S3
NFC	8.37 ^c^	6.52 ^d^	8.03	4.89 ^c^	5.32 ^d^	5.01	0.91	0.92	0.94
NFC_study_ ^a^	5.68	3.82	-	5.88	6.18	-	0.94	0.94	-
NFC_job_ ^b^	8.41	6.40	6.71	5.26	6.02	5.94	0.95	0.95	0.96
Positive affect	33.03	31.98	32.10	5.88	6.31	6.32	0.83	0.86	0.85
Negative affect	19.03	18.65	18.25	6.05	5.79	7.06	0.85	0.84	0.90
Life satisfaction	5.09	4.94	4.53	1.14	1.18	1.32	0.84	0.87	0.90
Job satisfaction ^b^	30.58	30.57	28.18	5.63	5.40	5.95	0.86	0.84	0.86
Study satisfaction ^a^									
Subject-related	74.94	72.70	-	20.21	19.90	-	0.92	0.90	-
Conditions	59.11	49.55	-	26.31	26.65	-	0.85	0.84	-
Coping with stress	62.20	59.62	-	26.70	24.23	-	0.87	0.84	-
Self-control	3.13	3.16	3.12	0.61	0.60	0.62	0.82	0.81	0.83
Reappraisal	4.70	4.45	4.59	1.04	0.95	1.01	0.83	0.78	0.85
Suppression	3.60	3.41	4.11	1.20	1.16	1.27	0.75	0.74	0.82

*Note. N =* 335 - 451. S1 = Sample 1. S2 = Sample 2. S3 = Sample 3. ^a^ *n* = 157/258. ^b^ *n* = 156–346. ^c^ Student subsample: *M* ± *SD* = 7.60 ± 4.80, working subsample: *M* ± *SD* = 9.29 ± 4.82. ^d^ Student subsample: *M* ± *SD* = 5.96 ± 5.26, working subsample: *M* ± *SD* = 7.12 ± 5.45.

**Table 4 jintelligence-12-00110-t004:** Correlations of NFC with general well-being, emotion regulation, and self-control.

		PA	NA	Life Satisfaction	Reappraisal	Suppression	Self-Control
NFC_general_	Sample 1	0.29 ***	−0.06	0.09	0.20 ***	0.11 *	0.25 ***
	Sample 2	0.31 ***	−0.02	0.14 *	0.16 **	0.04	0.11 *
	Sample 3	0.36 ***	−0.06	0.17 **	0.12 *	0.02	0.20 ***
NFC_study_	Sample 1	0.25 ***	−0.08	0.16 **	0.16 **	0.07	0.13 *
	Sample 2	0.25 **	−0.10	0.15	0.09	0.01	0.20 *
NFC_job_	Sample 1	0.39 ***	−0.12	0.21 **	0.22 **	0.01	0.27 ***
	Sample 2	0.28 ***	0.06	0.22 **	0.14	−0.02	0.03
	Sample 3	0.38 ***	−0.09	0.24 ***	0.13 *	−0.02	0.26 ***

*Note*. Spearman rank correlations. Sample 1: *N* = 451. Sample 2: *N* = 335. Sample 3: *N* = 346. Subsample sizes for correlations with NFC_study_ = 157/258, for NFC_job_ = 156–346. PA = positive affect. NA = negative affect. * *p* < 0.05. ** *p* < 0.01. *** *p* < 0.001 (Bonferroni-adjusted significance level).

**Table 5 jintelligence-12-00110-t005:** Correlations of NFC with domain-specific life satisfaction.

		Job Satisfaction	Study Satisfaction		
			Subject-Related	With Conditions	Coping with Stress
NFC_general_	Sample 1	0.11	0.31 ***	−0.03	0.07
	Sample 2	0.26 **	0.16 *	0.01	0.16 *
	Sample 3	0.20 ***	-	-	-
NFC_domain_	Sample 1	0.25 **	0.36 ***	0.01	0.12
	Sample 2	0.25 **	0.25 **	0.02	0.22 **
	Sample 3	0.33 ***	-	-	-

*Note*. Spearman rank correlations. Sample 1: *N* = 451. Sample 2: *N* = 355. Sample 3: *N* = 346. NFC_general_ = domain-general Need for Cognition. NFC_domain_ = study-specific Need for Cognition or job-specific Need for Cognition corresponding with domain of life satisfaction. Subsample sizes for correlations with NFC_study_ = 157/258, for NFC_job_ = 156–346. * *p* < 0.05. ** *p* < 0.01. *** *p* < 0.001 (Bonferroni-adjusted significance level).

**Table 6 jintelligence-12-00110-t006:** Hierarchical regression of NFC predicting positive affect.

	Student Sample 1 (*N* = 258)								
Predictor	Model 1			Model 2					
	*B*	*SE*	β	*B*	*SE*	β			
Constant	30.72 ***	0.68		30.81 ***	0.68				
NFC_general_	0.21 **	0.08	0.17	1.00	0.10	0.08			
NFC_study_				0.14	0.08	0.14			
*R*^2^/*R*^2^*_korr_*	0.030/0.026			0.040/0.033					
∆*R*^2^	0.030 **			0.010					
	Student Sample 2 (*N* = 157)								
Predictor	Model 1			Model 2			Model 3		
	*B*	*SE*	β	*B*	*SE*	β	*B*	*SE*	β
Constant	21.91 ***	3.30		19.99 ***	3.28		20.59 ***	3.35	
Age	0.38 **	0.14	0.19	0.36 *	0.14	0.18	0.34 *	0.14	0.17
NFC_general_				0.40 ***	0.10	0.32	0.28 *	0.13	0.22
NFC_study_							0.16	0.12	0.15
*R*^2^/*R*^2^*_korr_*	0.038/0.031			0.240/0.129			0.153/0.137		
∆*R*^2^	0.038 *			0.102 ***			0.013		
	Employed Sample 1 (*N* = 159)								
Predictor	Model 1			Model 2			Model 3		
	*B*	*SE*	β	*B*	*SE*	β	*B*	*SE*	β
Constant	30.50 ***	1.65		27.72 ***	1.71		26.91 ***	1.79	
Age	0.85 *	0.38	0.18	0.72 *	0.35	0.15	0.82 *	0.36	0.18
NFC_general_				0.36 ***	0.08	0.30	0.14	0.11	0.12
NFC_job_							0.29 **	0.11	0.27
*R*^2^/*R*^2^*_korr_*	0.033/0.027			0.125/0.114			0.162/0.146		
∆*R*^2^	0.033			0.092 ***			0.037 **		
	Employed Sample 2 (*N* = 156)								
Predictor	Model 1			Model 2			Model 3		
	*B*	*SE*	β	*B*	*SE*	β	*B*	*SE*	β
Constant	34.57 ***	0.76		31.89 ***	0.96		31.67 ***	0.95	
Gender ^a^	−2.30 *	0.95	−0.19	−1.85 *	0.91	−0.16	−1.69	0.89	−0.14
NFC_general_				0.34 ***	0.08	0.32	0.10	0.12	0.10
NFC_job_							0.28*	0.11	0.29
*R*^2^/*R*^2^*_korr_*	0.037/0.031			0.137/0.125			0.170/0.153		
∆*R*^2^	0.037 *			0.099 ***			0.033 *		
	Employed Sample 3 (*N* = 346)								
Predictor	Model 1			Model 2					
	*B*	*SE*	β	*B*	*SE*	β			
Constant	28.49 ***	0.60		28.83 ***	0.61				
NFC_general_	0.45 ***	0.06	0.36	0.18	0.12	0.15			
NFC_job_				0.27 **	0.10	0.25			
*R*^2^/*R*^2^*_korr_*	0.127/0.124			0.146/0.141					
∆*R*^2^	0.127 ***			0.019 **					

*Note*. NFC_general_ = domain-general Need for Cognition. NFC_study_ = study-specific Need for Cognition. NFC_job_ = job-specific Need for Cognition. SE and significance of coefficients based on 1000 Bootstrap samples. Variance inflation factor: 1.7–3.3. ^a^ 0 = male, 1 = female. * *p* < 0.05. ** *p* < 0.01. *** *p* < 0.001. Specific NFC explained 4% of variance in life satisfaction (*R*^2^*_marginal_* = 0.039).

**Table 7 jintelligence-12-00110-t007:** Hierarchical regression of NFC predicting life satisfaction.

	Student Sample 1 (*N* = 258)								
Predictor	Model 1			Model 2			Model 3		
	*B*	*SE*	β	*B*	*SE*	β	*B*	*SE*	β
Constant	6.39 ***	0.51		6.31 ***	0.54		6.31 ***	0.53	
Age	−0.46 *	0.18	−0.20	−0.45 *	0.18	−0.19	−0.44 *	0.18	−0.19
NFC_general_				0.01	0.01	.04	−0.02	0.02	−0.10
NFC_study_							0.04 *	0.02	0.20
*R*^2^/*R*^2^*_korr_*	0.038/0.034			0.039/0.032			0.061/0.050		
∆*R*^2^	0.038 **			0.001			0.022 *		
	Student Sample 2 (*N* = 157)								
Predictor	Model 1			Model 2			Model 3		
	*B*	*SE*	β	*B*	*SE*	β	*B*	*SE*	β
Constant	4.35 ***	0.21		4.09 ***	0.25		4.12 ***	0.25	
Gender ^a^	0.62 **	0.24	0.21	0.69 **	0.24	0.23	0.69 **	0.23	0.23
NFC_general_				0.03	0.02	0.15	−0.00	0.02	−0.02
NFC_study_							0.05 *	0.02	0.26
*R*^2^/*R*^2^*_korr_*	0.043/0.036			0.064/0.051			0.103/0.085		
∆*R*^2^	0.043 **			0.021			0.039 *		
	Employed Sample 1 (*N* = 159)								
Predictor	Model 1			Model 2			Model 3		
	*B*	*SE*	β	*B*	*SE*	β	*B*	*SE*	β
Constant	4.86 ***	0.30		4.75 ***	0.30		4.64 ***	0.31	
Age	0.10	0.07	0.12	0.09	0.07	0.11	0.11	0.07	0.13
NFC_general_				0.01	0.02	0.07	−0.02	0.03	−0.09
NFC_job_							0.04 *	0.02	0.22
*R*^2^/*R*^2^*_korr_*	0.14/0.008			0.019/0.006			0.044/0.025		
∆*R*^2^	0.014			0.004			0.025 *		
	Employed Sample 2 (*N* = 156)								
Predictor	Model 1			Model 2					
	*B*	*SE*	β	*B*	*SE*	β			
Constant	4.78 ***	0.19		4.76 ***	0.19				
NFC_general_	0.04	0.02	0.18	−0.00	0.03	−0.02			
NFC_job_				0.05 *	0.02	0.26			
*R*^2^/*R*^2^*_korr_*	0.032/0.026			0.059/0.046					
∆*R*^2^	0.032 *			0.027 *					
	Employed Sample 3 (*N* = 346)								
Predictor	Model 1			Model 2					
	*B*	*SE*	β	*B*	*SE*	β			
Constant	4.22 ***	0.14		4.30 ***	0.14				
NFC_general_	0.04 *	0.02	0.14	−0.02	0.03	−0.08			
NFC_job_				0.06 **	0.02	0.27			
*R*^2^/*R*^2^*_korr_*	0.020/0.017			0.042/0.036					
∆*R*^2^	0.020 **			0.021 **					

*Note*. NFC_general_ = domain-general Need for Cognition. NFC_study_ = study-specific Need for Cognition. NFC_job_ = job-specific Need for Cognition. SE and significance of coefficients based on 1000 Bootstrap samples. Variance inflation factor: 1.7–3.3. ^a^ 0 = male, 1 = female. *n* = 156 because of omitting one diverse participant for regression analyses. * *p* < 0.05. ** *p* < 0.01. *** *p* < 0.001.

**Table 8 jintelligence-12-00110-t008:** Hierarchical regression of NFC predicting study-related satisfaction.

	**Satisfaction with Study Subject**					
	Student Sample 1 (*N* = 258)					
Predictor	Model 1			Model 2		
	*B*	*SE*	β	*B*	*SE*	β
Constant	67.38 ***	2.56		68.28 ***	2.57	
NFC_general_	0.99 ***	0.28	0.24	−0.13	0.42	−0.03
NFC_study_				1.34 **	0.36	0.39
*R*^2^/*R*^2^*_korr_*	0.056/0.052			0.137/0.130		
∆*R*^2^	0.056 ***			0.082 ***		
	Student Sample 2 (*N* = 157)					
Predictor	Model 1			Model 2		
	*B*	*SE*	β	*B*	*SE*	β
Constant	69.97 ***	2.43		70.71 ***	2.35	
NFC_general_	0.46	0.30	0.12	−0.39	0.44	−0.10
NFC_study_				1.12 **	0.38	0.35
*R*^2^/*R*^2^*_korr_*	0.015/0.008			0.086/0.074		
∆*R*^2^	0.015			0.071 ***		
	**Coping with Study-Related Stress**					
	Student Sample 2 (*N* = 157)					
Predictor	Model 1			Model 2		
	*B*	*SE*	β	*B*	*SE*	β
Constant	55.97 ***	3.02		56.58 ***	2.93	
NFC_general_	0.61	0.40	0.13	−0.08	0.55	−0.02
NFC_study_				0.93 *	0.47	0.24
*R*^2^/*R*^2^*_korr_*	0.018/0.011			0.051/0.038		
∆*R*^2^	0.018			0.033 *		

*Note*. NFC_general_ = domain-general Need for Cognition. NFC_study_ = study-specific Need for Cognition. Variance inflation factor: 1.7–2.1. *SE* and significance of coefficients based on 1000 Bootstrap samples. * *p* < 0.05. ** *p* < 0.01. *** *p* < 0.001.

**Table 9 jintelligence-12-00110-t009:** Hierarchical regression of NFC predicting job satisfaction.

	Employed Sample 1 (*N* = 159)								
Predictor	Model 1			Model 2			Model 3		
	*B*	*SE*	β	*B*	*SE*	β	*B*	*SE*	β
Constant	27.21 ***	1.62		27.29 ***	1.75		26.15 ***	1.74	
Age	0.80 *	0.37	0.17	0.80 *	0.37	0.17	0.94 *	0.37	0.20
NFC_general_				−0.01	0.10	−0.01	−0.32 **	0.11	−0.27
NFC_job_							0.41 ***	0.09	0.38
*R*^2^/*R*^2^*_korr_*	0.029/0.023			0.029/0.016			0.103/0.085		
∆*R*^2^	0.029*			0.000			0.074 ***		
	Employed Sample 2 (*N* = 156)								
Predictor	Model 1			Model 2					
	*B*	*SE*	β	*B*	*SE*	β			
Constant	28.62 ***	0.78		28.58 ***	0.78				
NFC_general_	0.27 **	0.08	0.28	0.17	0.12	0.18			
NFC_job_				0.12	0.12	0.15			
*R*^2^/*R*^2^*_korr_*	0.077/0.071			0.084/0.072					
∆*R*^2^	0.077 ***			0.007					
	Employed Sample 3 (*N* = 346)								
Predictor	Model 1			Model 2					
	*B*	*SE*	β	*B*	*SE*	β			
Constant	26.50 ***	0.64		27.10 ***	0.62				
NFC_general_	0.21 **	0.07	0.18	−0.27 *	012	−0.23			
NFC_job_				0.48 ***	0.10	0.48			
*R*^2^/*R*^2^*_korr_*	0.031/0.028			0.099/0.094					
∆*R*^2^	0.031 **			0.068 ***					

*Note*. NFC_general_ = domain-general Need for Cognition. NFC_job_ = job-specific Need for Cognition. SE and significance of coefficients based on 1000 Bootstrap samples. Variance inflation factor: 1.9–3.3. * *p* < 0.05. ** *p* < 0.01. *** *p* < 0.001.

**Table 10 jintelligence-12-00110-t010:** Mediation analyses with self-regulatory processes as parallel mediators.

Predictor	Sample	Indirect Effects				Direct ^b^
		Total ^a^	SCS ^b^	Reappraisal ^b^	Suppression ^b^	
Positive Affect						
NFC_general_	Sample 1	0.10 [0.05, 0.14] *	x	x	x	x
	Sample 2	0.06 [0.01, 0.11] *	ns	x	ns	x
	Sample 3	0.09 [0.04, 0.14] *	x	x	ns	x
NFC_study_	Sample 1	0.07 [0.01, 0.13] *	x	x	ns	x
	Sample 2	0.07 [−0.01, 0.16]	x	ns	ns	x
NFC_job_	Sample 1	0.13 [0.05, 23] *	x	ns	ns	x
	Sample 2	0.06 [−0.02, 0.15]	ns	x	ns	x
	Sample 3	0.11 [0.06, 0.18] *	x	x	ns	x
Life Satisfaction						
NFC_general_	Sample 1	0.10 [0.06, 0.15] *	x	x	x	ns
	Sample 2	0.05 [−0.01, 0.09]	ns	x	ns	x
	Sample 3	0.08 [0.03, 0.12] *	x	x	ns	ns
NFC_study_	Sample 1	0.07 [0.02, 0.13] *	x	x	ns	ns
	Sample 2	0.09 [0.01, 0.19] *	x	ns	ns	ns
NFC_job_	Sample 1	0.12 [0.04, 0.22] *	x	ns	ns	ns
	Sample 2	0.03 [−0.03, 0.10]	ns	ns	ns	x
	Sample 3	0.09 [0.04, 0.14] *	x	x	ns	x
Study-Related Satisfaction: Subject-Related						
NFC_general_	Sample 1	0.01 [−0.03, 0.06]	ns	ns	ns	ns
	Sample 2	0.05 [−0.02, 0.12]	ns	ns	ns	ns
NFC_study_	Sample 1	0.04 [−0.01, 0.09]	ns	x	ns	x
	Sample 2	0.07 [0.01, 0.14] *	x	ns	ns	x
Study-Related Satisfaction: Coping						
NFC_general_	Sample 1	0.01 [−0.04, 0.06]	ns	ns	ns	ns
	Sample 2	0.04 [−0.01, 0.12]	ns	ns	ns	ns
NFC_study_	Sample 1	0.01 [−0.03, 0.05]	ns	ns	ns	ns
	Sample 2	0.04 [−0.02, 0.12]	ns	ns	ns	x
Job Satisfaction						
NFC_general_	Sample 1	0.05 [0.00, 0.12] *	ns	x	ns	x
	Sample 2	0.01 [−0.04, 0.07]	ns	ns	ns	x
	Sample 3	0.03 [−0.00, 0.07]	x	ns	ns	x
NFC_job_	Sample 1	0.06 [0.00, 0.13] *	x	ns	ns	ns
	Sample 2	0.01 [−0.04, 0.08]	ns	ns	ns	x
	Sample 3	0.04 [0.01, 0.08] *	x	ns	ns	x

*Note*. *N* = 335–451. BCa 95% confidence intervals of 1000 Bootstrap samples. Significance of effects based on confidence intervals excluding 0. ^a^ Completely standardized. ^b^ x = significant effect. ns = non-significant. * Confidence interval excluding 0.

## Data Availability

The original data presented in this study are openly available in https://doi.org/10.17605/osf.io/y6evn.

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
