# Peer review of "Cognitive Engagement and Subjective Well-Being in Adults: Exploring the Role of Domain-Specific Need for Cognition"

_jintelligence, 2024, doi:10.3390/jintelligence12110110_

Round 1
Reviewer 1 Report
Comments and Suggestions for Authors
The manuscript "Cognitive Engagement and Subjective Well-Being in Adults: 2 Exploring the Role of Domain-Specific Need for Cognition" is well-written and a well-powered study.
The authors dived into the connection between well-being and domain-specific Need for Cognition inspired by findings in children (Keller et al.) and the review by Zerna et al that domain-general NFC is positively associated with well-being.
A strong asset is the pre-registration and provision of raw data. I like that very much (although the data is not in FAIR i.e. *.sav but that is just a minor issue).
Two analysis-related questions.
The authors choose to run a null model with age and gender and if one of them is not significant, leave it out when entering the other variables (e.g. NFC general, NFC domain). However, this assumes no moderation effect of age and gender, but there are studies showing age effects for NFC. The models for the three studies could also be compared if age and gender is always included, i.e. the beta's can be compared and a meta-analysis run.
The second point is why the authors have not run an analysis with collated data or as linear mixed model with data nested within the three waves and nested within student / employee. This would allow calculating an overall effect of NFC general / NFC domain on well-being - separate for students and employees. The rational why this is not done is not well described in the manuscript.
The samples may differ (corona) but that does not imply that the association between NFC and positive affect changes (if all rate their well-being by one "well-being unit" lower during the pandemic, the correlation remains the same
It seems that NFC domain general was always measured before NFC domain specific, or was that counterbalanced? Is the e.g. 8.37 and 5.68 the sum score? The NFC is rated on a 7-point Likert scale, from -3 to +3, since you report the sum score, the 8.37 refers to on average rating all five items as "agree" whereas for e.g., 3.82 this is not the case. So students do not agree on thinking hard for their study topic - I find in general the discrepancy between general vs domain for students surprising. They rate themselves as more liking thinking but not liking to think study-specific. Any idea why that is? The gap for employees is not so large (see particularly S2)
minor issues
line 151: space missing between "(H3).For"
line 223: twice "the because"
line 239: "was small associated" should be "was weakly associated"
statistically one can say "predicted" but since you cannot infer causality, I recommend to replace "predicted" with related or associated. We do not know whether life satisfaction (more a state than a trait) predicts NFC (more a trait) or vice versa. Likely it is an interaction, not least for domain-specific NFC. The puzzling reduction in students might also be driven by the adverse study-conditions (home instead on campus).
A more general weakness is that the students / employees were not ask about a non-domain NFC as this would strengthen the "modularity" argument of domain general vs domain-specific aspects.
This manuscript is based in self-rated questionnaires that are not counterbalanced. Should be mentioned as limitations.
Overall an interesting project and I enjoyed reading the manuscript.
Author Response
Please see the attachment. For better understanding all changes during the revision process, we upload one response letter for both reviewers.

Reviewer 2 Report
Comments and Suggestions for Authors
The present study investigates in three samples (1) whether domain-specific need for cognition (NFC) incrementally predicts subjective well-being (SWB) above and beyond domain-general NFC; and (2) whether self-control and emotion regulation strategies mediate the link between NFC and SWB. The authors find some support that domain-specific NFC provides an incremental predictive value for positive affect and (domain-general and domain-specific) life satisfaction. The authors also find some evidence for a mediation effect by reappraisal and self-control.
Although not related to intelligence, the paper has a good fit to the special issue related to cognitive motivation. I have some comments on the current version of the paper that I will outline in the following.
The Introduction is well-structured. However, while paragraph 1.2 is thought to focus on the theoretical explanation why there should be a relation between NCF and SWB, I did not find much information really focusing on this question. It would be great if the authors could explain more clearly why such a relation might be theoretically expected and why self-control and emotion regulation strategies can be expected to mediate this causal relation. Strengthening this theoretical background would be especially important given that all analyses were correlational and cross-sectional. Thus, an additional focus might be on a potential reversed causation.
I was wondering about multicollinearity. I would expect domain-specific and domain-general NFC to be highly correlated. Might this affect the results from the regression models? It is remarkable that the regression weights of general NFC often turned negative when domain-specific NFC was entered into the model (see Tables 7 and 8). A correlation table of the study variables is missing.
In general, the authors performed lots of significance tests but did not correct for alpha error inflation.
Why was gender not dummy-coded for the regression analyses?
Table 1: What does the abbreviation MD mean? And why is it 21-29 years for overall sample 1 and 30-39 years for the employees in sample 1?
There is a mismatch between the student subsample 1 in Table 1 (n = 258) and in the text (n = 259; l. 212).
Please explain BCa (introduce abbreviation).
It should be noted in Tables 4 and 5 (either in the table headers or notes) that the correlations are Spearman correlations. This should also be clear in the running text, where the authors use “r” instead of “ρ,” which is misleading.
Please explain “PA” and “NA” in the table note of Table 4.
In the header of Table 5 it should read “with Domain-Specific Life Satisfaction.” Life satisfaction is part of SWB and this table only presents results for life satisfaction. The same is true for the running text (see l. 340).
The cross-sectional correlational design is a strong limitation of the study, but the authors do not mention it as such. What about the possibility of reversed causation? It is also not optimal to use cross-sectional data for mediation analyses.
I was also unsure why the items for domain-specific NFC were seen by the authors as a limitation. Please explain.
Multicollinearity might also be a limitation that warrants discussion.
Finally, self-selection might be a problem not just for generalizability but also regarding variance restriction. Might it be possible that individuals high in NFC and in SWB were more likely to take part in the study than individuals low in NFC and SWB?
Line comments:
- l. 53: It is “range from … to …”
- l. 69: “in the long run”
- l. 96: I did not understand “for predicting well-being associated indicators by NFC”. Do you mean “predict from NFC?” And why not just say “well-being indicators”?
- l. 178: “referred”
- l. 205: The first dataset was collected between … and …” (or “from … to …”)
- l. 223: Delete “because the”
- l. 227: “correlated strongly with” (see also l. 516)
- ll. 474-476: Consider rephrasing (e.g., “Whereas study-specific NFC had no incremental predictive value over and above domain-general NFC, job-specific NFC explained some incremental variance”)
- l. 480: Delete comma
- ll. 488-489: “moderate to strong associations of both job satisfaction and study-subject related satisfaction with global life satisfaction”
- ll. 498-499: “during the COVID-19 pandemic”
- ll. 342, 526-527: Expressions such as “small related” are wrong English. Please consider revising (e.g., “weakly related” or the like)
- l. 537: “(emotional) challenges” … “self-regulatory processes”
- l. 589: Insert comma after “Further”
Comments on the Quality of English LanguageSee line comments.
Author Response

(The authors gave the same response as above.)

Round 2
Reviewer 1 Report
Comments and Suggestions for Authors
the authors addressed all my comments.
The new analysis are attached as appendix and are on the osf page
Author Response
Comment 1: The authors addressed all my comments.
The new analysis are attached as appendix and are on the osf page
Response 1:
Thank you very much for providing your review so fast and the positive evaluation of our revision.
Reviewer 2 Report
Comments and Suggestions for Authors
The authors have completed a major revision of their paper and as a result, the paper has improved. I only have two comments left:
I found the sentence running from ll. 371-374 quite long, a bit awkward and therefore hard to understand. I would recommend rephrasing and maybe making two sentences of it.
In the notes of Tables 4 and 5, the authors noted Bonferroni-correction directly following the p-values. This seems a bit misleading, because critical alpha (and not p-values) can be corrected. Instead, it would be better to note that critical alpha equaled .001 after correction (or maybe something like "*** significant after Bonferroni correction").
Comments on the Quality of English LanguageThere are minor language mistakes throughout the paper, but they do not undermine comprehensibility in my view.
Author Response
Reviewer 2:
Comment 1:
The authors have completed a major revision of their paper and as a result, the paper has improved. I only have two comments left.
Response 1:
Thank you very much for providing your review so fast and the positive evaluation of our revision.
Comment 2:
I found the sentence running from ll. 371-374 quite long, a bit awkward and therefore hard to understand. I would recommend rephrasing and maybe making two sentences of it.
Response 2:
We corrected the sentence as follows:
Mediation analyses were calculated for the prediction of all well-being indicators NFC was associated with. Analyses were done with PROCESS (version 4.0; Hayes, 2022) calculating 95% BCa confidence intervals based on 1000 bootstrap samples.
Comment 3:
In the notes of Tables 4 and 5, the authors noted Bonferroni-correction directly following the p-values. This seems a bit misleading, because critical alpha (and not p-values) can be corrected. Instead, it would be better to note that critical alpha equaled .001 after correction (or maybe something like "*** significant after Bonferroni correction").
Response 3:
We completely agree and modified the note as follows: *** p < .001 (Bonferroni-adjusted significance level).
There was a mistake in the following sentence referring to the significance of (in)direct effects, which was corrected.
Comment 4:
There are minor language mistakes throughout the paper, but they do not undermine comprehensibility in my view.
Response: 4: we went through the manuscript and shortened or reworded some sentences.
Please find all changes with red markup in the revised manuscript.
Round 3
Reviewer 2 Report
Comments and Suggestions for Authors
The authors have resolved the remaining issues. I have no comments left.